# Improving the Flavour of Enzymatically Hydrolysed Beef Liquid by Sonication

**DOI:** 10.3390/foods12244460

**Published:** 2023-12-13

**Authors:** Chao Ye, Zhankai Zhang, Zhi-Hong Zhang, Ronghai He, Xue Zhao, Xianli Gao

**Affiliations:** School of Food and Biological Engineering, Jiangsu University, Zhenjiang 212013, China; 2222118044@stmail.ujs.edu.cn (C.Y.); 2212218064@stmail.ujs.edu.cn (Z.Z.); zhihong1942@ujs.edu.cn (Z.-H.Z.); heronghai1971@126.com (R.H.); 2222118035@stmail.ujs.edu.cn (X.Z.)

**Keywords:** beef potentiator, enzymatically hydrolysed beef liquid, ultrasound, aldehydes, alcohols, flavour

## Abstract

Beef potentiator is an important flavour enhancer in the food industry, while it is prone to generating insufficient compounds with umami and sweet tastes and compounds with a fishy odour during enzymatic hydrolysis of beef, resulting in poor flavour of beef potentiator. It has been extensively reported that sonication is capable of improving food flavour. However, the effect of sonication on the flavour of enzymatically hydrolysed beef liquid (EHBL) was scarcely reported. Herein, we investigated the effect of sonication on the flavour of EHBL using quantitative descriptive analysis (QDA), physicochemical analysis and SPME-GC-olfactometry/MS. QDA showed that sonication had a significant effect on taste improvement and off-odour removal of EHBL. Compared with the control, sonication (40 kHz, 80 W/L) increased the contents of total nitrogen, formaldehyde nitrogen, total sugars, reducing sugars, free amino acids (FAAs) and hydrolysis degree of EHBL by 19.25%, 19.80%, 11.83%, 9.52%, 14.37% and 20.45%. Notably, sonication markedly enhanced the contents of sweet FAAs, umami FAAs and bitter FAAs of EHBL by 19.66%, 14.04% and 9.18%, respectively, which contributed to the taste improvement of EHBL. SPME-GC-olfactometry/MS analysis showed that aldehydes and alcohols were the main contributors to aroma compounds of EHBL, and sonication significantly increased the contents of key aroma compounds and alcohols (115.88%) in EHBL. Notably, sonication decreased the contents of fishy odorants, hexanoic acid and nonanal markedly by 35.29% and 26.03%, which was responsible for the aroma improvement of EHBL. Therefore, sonication could become a new potential tool to improve the flavour of EHBL.

## 1. Introduction

Thermal reaction flavour is a kind of food flavour enhancer and plays an important role in the modern food industry; it originated in European countries in the 1970s and is prepared via the Maillard reaction using enzymatically hydrolysed animal protein and reducing sugars as materials [1]. Nowadays, the annual output of thermal reaction flavour in European and American countries has exceeded 30,000 tons, accounting for more than 65% of the total flavour output, of which beef potentiator is the most popular thermal reaction flavour due to its popular flavour. Undoubtedly, beef potentiator will still have a huge demand in the future market, because the flavour preference of consumers is difficult to change. Beef potentiator was prepared by mincing beef, grinding beef, boiling beef, enzymatic hydrolysis (protease, papain and flavourzyme are usually used), adding reducing sugar and amino acids and the Maillard reaction at 110–130 °C, then auxiliary materials (salt, monosodium glutamate, maltodextrin, preservative, etc.) were added to the Maillard reaction product at the end of the Maillard reaction, followed by spray drying and packaging [2]. The main production process is shown in Figure 1. The enzymatic hydrolysis of beef is generally regarded as the most important process to obtain a beef potentiator with palatable flavour, because the free amino acids and peptides with low molecular weights generated in the process affect the flavour of beef potentiator significantly [3].

However, it is still difficult to obtain enzymatically hydrolysed beef liquid (EHBL) with high contents of free amino acids and peptides with low molecular weight due to limitations of the activity and cost of enzymes and the processing conditions used. Furthermore, the off-odour (i.e., fishy odour) usually occurs during enzymatic hydrolysis of beef, which seriously lowers the quality of beef potentiator. Previous investigations demonstrated that beef aroma was associated with 38 aroma compounds [4], but the odour compounds relating to the off-odour of EHBL were still not clear till now. Thus, it is necessary to explore new methods to improve the flavour of EHBL.

Ultrasound is an effective method to improve the enzymatic hydrolysis of protein and the physicochemical properties of enzymatically hydrolysed products [5]. Appropriate low-intensity ultrasound can increase the hydrolysis degree and improve the flavour of hydrolysed products [6]. At present, ultrasound has been applied in the manufacturing of various foods to enhance food properties and/or productivity. For example, Hu et al. found that the combination of ultrasonic treatment (300 W/30 kHz/30 min) and fermentation with *Bifidobacterium lactis* BP2 enhanced the taste and odour characteristics of beef jerky and improved its tenderness and juiciness [7]. In addition, the combination of ultrasound and protease showed positive effect on beef tenderization [8]. Yang et al. treated grass carp protein during enzymatic hydrolysis using energy-divergent ultrasound with a power of 100 W/L for 20 min at 30 ± 2 °C. The results showed that content of free amino acids in the sonicated hydrolysate reached 242.83 mg/mL, which was 30.96% higher than that of the control (185.43 mg/mL); the taste of the sample was improved significantly, and its bitterness and caramel colour decreased markedly (*p* < 0.05); finally, the overall acceptability of the sample was remarkably enhanced [9]. Bai et al. also found that ultrasonic treatment (300 W, 60 min) had a positive impact on the texture and flavour of pickled bass [10]. In addition, previous investigation demonstrated that sonication (68 kHz, 60 W/L, 10 min, eight circles) during soy sauce fermentation significantly enhanced the level of total FAAs in samples in comparison to the controls, suggesting that sonication promoted the amino acids release and accelerated flavour maturation of soy sauce [11]. The cavitation, mechanical and chemical effects of ultrasound may increase the hydrolysis degree of substrates by altering the structures of enzymes and substrates, increasing the specific surface area of substrates and their collision probability, thereby promoting the release of flavour substances such as free amino acids, small molecule peptides, nitrogen-containing substances and reducing sugars [12,13,14,15]. But it is scarcely reported whether sonication could improve the flavour of EHBL.

Therefore, the objectives of this study were to investigate (i) the effect of sonication on the flavour of EHBL and (ii) to further explore the effect of sonication on the taste compounds and aroma compounds (especially off-aroma compounds) of EHBL.

## 2. Materials and Methods

### 2.1. Materials and Chemicals

Beef was commercially available from a local supermarket (Jimailong supermarket, Zhenjiang, China). Flavourzyme^®^ 500 MG (EC3.4.11.1, complex of exoprotease and endoprotease from *Aspergillus oryzae*) and Neutrase 0.8 L were purchased from Novozymes Biotechnology Co., Ltd. (Beijing, China). Hexanoic acid, nonanal, 2-pentylfuran, (E)-2-octenal, (2E)-2-nonenal, (E,E)-2,4-decadien-1-al, 1-octen-3-ol and C6-C33 n-alkanes were provided by Aladdin Holdings Group (Shanghai, China, 2021). Other reagents with analytical purity were purchased from China National Pharmaceutical Group Chemical Reagent Co., Ltd. (Shanghai, China).

### 2.2. Beef Homogenate Preparation and Treatment

Preparation of beef homogenate: The beef was mixed with distilled water at a ratio of 1:1 (*w*/*w*); the mixture was steamed at 100 °C for 1 h; then, the beef was cut into small pieces and ground through a colloid mill to make beef homogenate. Then, the beef homogenate was evenly divided into 6 aliquots; 3 aliquots were used to prepare the control, and the other 3 aliquots were used to prepare the sample.

Control: Firstly, Neutrase 0.8 L was added into the beef homogenates (300 U/g beef homogenate), followed by incubating the homogenates at 60 °C in a water bath for 1 h; then, Flavourzyme^®^ 500 MG (300 U/g beef homogenate) was added into the aforementioned beef homogenates, which were enzymatically hydrolysed for another 2 h at 60 °C. The pH of the homogenates was controlled at about 6.5 ± 0.5 by adding 2 M NaOH solution. The selection of enzymes used in this work was based upon Gao et al. [1,2].

Sample: The homogenates were treated as the above except sonication being conducted during enzymatic hydrolysis. Sonication (40 kHz, 80 W/L, 10 s on/10 s off; sonication conditions were optimized previously) was performed immediately after adding Neutrase 0.8 L.

### 2.3. Sensory Evaluation

QDA was performed to investigate the differences in taste and aroma between the sample and control [9,16]. QDA was carried out with a panel of seven evaluators (4 males and 3 females, aged 24–45) majoring in food science and technology at Jiangsu University, Zhenjiang, China. Prior to training, the evaluators were asked to rank a series of 10-fold suprathreshold aqueous solutions (25 mL in Teflon vials) of ethanol (alcoholic), acetic acid (sour), nonanal (fatty), trimethylpyrazine (roast), bis (2-methyl-3-furyl) disulfide (meat-like) and trimethylamine (fishy) during aroma evaluation. The following five reference solutions were prepared to recalibrate the five basic tastes during taste evaluation: 4 mM monosodium glutamate (pH 5.6, umami), 40 mM saccharose (sweet), 12 mM NaCl (salty), 1.5 mM caffeine (bitter) and 10 mM lactic acid (sour). Then, the evaluators were asked to score the taste and aroma intensities on a linear scale from 0 to 9. The experiments were carried out in a laboratory at 23 ± 2 °C; the results of the sensory evaluation were discussed and agreement was eventually reached by all the evaluators. Sensory analysis was repeated three times.

### 2.4. Analyses of Total Nitrogen, Formaldehyde Nitrogen, Hydrolysis Degree, Total Sugars and Reducing Sugars

Prior to analysis, each sample and control were filtered with a qualitative filter paper. Total nitrogen content was determined by the Kjeldahl method (AOAC 920.87, 1995), and formaldehyde nitrogen was determined by titration [16]. The hydrolysis degree of EHBL was measured based on the pH-stat method described by Rezvankhah et al. [17]. Total sugars were estimated by the phenol–sulfuric acid method [18]. The reducing sugars were assessed using 3,5-dinitrosalicylic acid titration [19].

### 2.5. Free Amino Acids Analysis

FAAs were estimated according to the method described by Gao et al. with minor modifications [16]. The control and sample were diluted with 10% trichloroacetic acid solution; the mixtures were left to stand for 2 h, then were filtered through double filter paper. Then, 1 mL of the filtrate was placed in a 1.5 mL centrifuge tube and centrifuged at 10,000× *g* for 30 min. The supernatant was filtered through a 0.45 μm pore filter, and the filtrate was subjected to high-performance liquid chromatography (Waters Ltd., Milford, MA, USA) with a PICO.TAG amino acid analysis column (3.9 mm i.d. × 150 mm length). Ten microlitres of the filtrate was injected into the column, then was eluted with at 1.0 mL/min and monitored at 254 nm. Elution was carried out by using a gradient procedure with a mobile phase containing solvent A and solvent B as follows: 0 min, 5% B; 15 min, 20% B; 35 min, 40% B; 42 min, 65% B; 50 min, 80% B; 52 min, 5% B; 60 min, 5% B; the run time was 60 min, the solvent flow rate was 1.0 mL⁄ min and the injection volume was 10 uL. Solvent A comprised 940 mL of 0.14 M sodium acetate (pH 6.40, containing 0.05% triethylamine) and 60 mL of acetonitrile. Solvent B was 60% acetonitrile and 40% water by volume. The FAA concentrations in the sample and control were quantitated using amino acids standard solution (AAS18; Sigma, St. Louis, MO, USA).

### 2.6. Thresholds and Taste Intensity of FAAs

The thresholds of FAAs in water were independently investigated in this study, and the selection and training of the panellists were described in Section 2.3. The lowest concentration of each FAA perceived by more than half of nine panellists was regarded as its taste threshold. The taste intensity of each FAA was determined using the following equation:(1)$$\mathrm{Taste}\;\mathrm{intensity}=\frac{FAA\;concentration}{FAA\;threshold\;in\;water}$$

### 2.7. Protease Activity Analysis during Sonication

During the enzymatic hydrolysis of beef homogenate, EHBL was taken (0 min, 30 min, 60 min, 90 min, 120 min, 150 min and 180 min) to determine the neutral protease activities of the control and sample. The activity of neutral protease in EHBL was determined using the method described by Gao et al. [11]. The specific operations were as follows: 2% casein (*w*/*v*) dissolved in 100 mM sodium phosphate buffer at pH 7.2 was utilized as the substrate for the determination of neutral protease activity. The filtrate of EHBL was diluted to a proper concentration using the sodium phosphate buffer. One millilitre of the diluted filtrate and 1 mL of its corresponding substrate were mixed and incubated for 10 min at 40 °C. Then, 2 mL of trichloroacetic acid (5%) was added to settle the unhydrolysed protein, which was removed using Whatman No.1 filter paper. One millilitre of the resulting filtrate was thoroughly mixed with 5 mL of sodium bicarbonate (0.4 M) for the following reaction with 1 mL of Folin–Ciocalteu phenol reagent (200 mM) for 20 min. After centrifugation at 10,000× *g*, the absorbance value of the resulting supernatant was monitored at 660 nm.

### 2.8. Volatile Compound Extraction Using Solid-Phase Micro-Extraction (SPME)

To investigate the effect of sonication on EHBL odour, the sample (10 mL) and control (10 mL) were put into 20 mL headspace bottles, and the volatile compounds were extracted from the control and sample using a SPME fibre coated with carboxypoly dimethylsiloxane (Supelco, Bellefonte, PA, USA). Before extraction, the sample was preheated for 30 min at 45 °C in a headspace bottle and stirred continuously with a magnetic stirrer. Then, volatile compounds were adsorbed using SPME fibre at 45 °C for 30 min. The fibre was directly introduced into a GC injector for 5 min at 250 °C and then further separated by GC-MS. In all cases, the fibres were conditioned before use by inserting them into the GC injector port for 1 h at 260 °C and then were desorbed for 10 min at 260 °C between injections to prevent any contamination.

### 2.9. Analyses of GC-Olfactometry and Flavour Dilution Factor

As previously described by Gao et al. [20], the analyses of GC-Olfactometry (GC-O) were conducted using an Agilent 6890 gas chromatograph equipped with an Agilent 5973 N mass selection detector (Wilmington, DE, USA) and a sniffing port (ODP-2; Gerstel, Inc., Linthicum, MD, USA). The collection of volatile compounds, GC columns and conditions were the same as described in Section 2.10.

The FD factor of each volatile compound represents the maximum dilution that the odour can be perceived at the olfactory mouth by two-thirds of the evaluators [21,22,23]. In this study, the dilution of the volatile compound was represented by the split ratios of GC injection, which were 1:1, 2:1, 4:1, 8:1, 16:1, 32:1, 64:1 and 128:1.

### 2.10. Qualitative Analysis of Volatile Compounds

Qualitative analysis of volatile compounds was performed with GC-MS system and a DB-Wax column (30 m × 0.25 mm × 0.25 μm; J&W Science, Folsom, CA, USA). High-purity helium (99.999%) was used as carrier gas with a flow rate of 1.0 mL/min (constant flow mode). The column was maintained for 3 min at 40 °C, heated from 40 °C to 200 °C with a speed of 5 °C/min and heated from 200 °C to 230 °C with a speed of 10 °C/min. The mass spectrum conditions used were as follows: the ion source temperature was 230 °C, the electron energy was 70 eV and the mass scanning range was 30–450 *m*/*z*. Volatile compounds were identified according to the Kovats retention index (RI) and by matching mass spectrometry in the NIST05 library (Gaithersburg, MD, USA). In addition, RIs were determined under the same GC conditions using C6-C33 n-alkanes as the standards [24].

### 2.11. Quantitative Analysis of Key Odorants

The external standard method was used to quantify the odour-active compounds in EHBL. The peak area of each odour-active compound was compared with peak area of its external standard compound to estimate its concentration. The response factor of each odour-active compound to its external standard compound was regarded as 1 [23].

### 2.12. Odour Activity Value (OAV)

OAV is the ratio between the concentration of an odour compound and its odour threshold value detected in water. The contribution of an odour compound to EHBL was evaluated by its OAV; the contribution of an odorant with OAV beyond 1 was regarded as an odour-active compound to EHBL [23].

### 2.13. Statistical Analysis

In this study, all tests except sensory tests were repeated thrice, and all data were expressed as mean ± standard deviation (SD). SPSS 15.0 software (SPSS Inc., Chicago, IL, USA) was used for one-way ANOVA to determine that the differences were statistically significant within a 95% confidence interval, and the significance level was set to *p* < 0.05.

## 3. Results and Discussion

### 3.1. Sensory Evaluation

As shown in Figure 2, sensory analysis is the most direct and effective method to evaluate food flavour. In this work, the effect of sonication on the flavour of EHBL was evaluated by QDA. As shown in Figure 2A, compared with the control, the sensory evaluation scores of umami, sweet and bitter tastes of the sample were 17.24%, 15.79% and 9.93% higher than those of the control. One-way ANOVA demonstrated that the sensory evaluation scores of umami and sweet tastes of the sample were significantly higher than those of the control (*p* < 0.05), indicating that sonication during the enzymatic hydrolysis improved the taste of EHBL markedly. This is consistent with the results of free amino acid testing later, and ultrasound can further promote the release of taste amino acids (umami and sweet).

As shown in Figure 2B, compared with the control, the sensory evaluation scores of fishy and fatty aromas of the sample were 39.62% and 32.73% lower than those of the control; one-way ANOVA demonstrated that the sensory evaluation scores of both aromas of the sample were significantly reduced compared to those of the control (*p* < 0.05). Furthermore, sonication enhanced roast and alcoholic aromas of the sample to some extent. The above results indicated that sonication during the enzymatic hydrolysis improved the aroma of EHBL significantly. Previous investigations have extensively explored the effect of sonication on the flavour of fermented foods and enzymatically hydrolysed products; the results showed that moderate sonication could improve their taste and aroma [11,20,25], which was in line with the present sensory evaluation results. Overall, QDA proved that moderate sonication had a positive effect on the improvement of EHBL flavour, the decreases in fishy and fatty flavours may be related to ultrasound altering the release of free fatty acids in beef homogenate and the increase in baking flavour may also be related to the energy brought by ultrasound [26]. However, the effect of sonication on the specific taste and aroma compounds of EHBL needs further exploration.

### 3.2. Analyses of Total Nitrogen, Formaldehyde Nitrogen, Hydrolysis Degree, Total Sugars and Reducing Sugars

The release of nutrients in beef, such as proteins and sugars, plays an important role in the formation of flavour in the final product. Under the action of sonication, proteins and sugars can further degrade into flavour substances or flavour precursors, significantly increasing the content of volatile flavour substances in the final product. Therefore, it is necessary to measure the following indicators. As shown in Table 1, compared with the control, contents of total nitrogen (10.53 g/L), formaldehyde nitrogen (1.21 g/L), hydrolysis degree (19.67%), total sugars (1.04 g/L) and reducing sugars (0.92 g/L) of the sample were 19.25%, 19.80%, 20.45%, 11.83% and 9.52% higher than those of the control, indicating that sonication (40 kHz, 80 W/L, 10 s on/10 s off) not only enhanced the hydrolysis degree of beef and the contents of sugars and formaldehyde nitrogen, but also promoted solubility of N-containing compounds (i.e., proteins and peptides) and carbohydrates of beef (*p* < 0.05). Previous research indicated that appropriate sonication could increase 33–106% of the protein degradation rate by proteases and enhance the contents of hydrolysed products of protein [27,28], which was consistent with our present results.

Previous research demonstrated that sonication changed the spatial structure of enzymes and substrates, which enlarged the specific area of the substrate and increased the contact chance of protease and the substrate, and then enhanced the protease activity and hydrolysis degree and contents of taste compounds [11], which could also explain the above phenomena observed in our experiment.

### 3.3. Analysis of Free Amino Acids

Sonication accelerated the release of FAAs during meat processing [29,30]. FAAs, especially glutamic acid (an amino acid with an umami flavour), were regarded as key contributors to taste of food [11,25]. Compared with the control, the total FAA level in the sample increased by approximately 14.37%, which was similar to the increase in formaldehyde nitrogen in the sample (19.80%). Specifically, sonication increased the levels of most sweet FAAs (serine, glycine, threonine, alanine, proline and lysine), bitter FAAs (histidine, arginine, valine, tyrosine, methionine, leucine, isoleucine and phenylalanine) and umami FAAs (aspartic acid and glutamic acid) by 19.66%, 9.18% and 14.04%. Correspondingly, sonication increased the sweetness intensity, bitterness intensity and umami intensity of EHBL by 15.79%, 9.93% and 17.24% in the taste evaluation. In addition, it was found that sonication slightly reduced the glutamate level. Gao et al. found that low-intensity sonication (60 W/L, 10 min × 8) during soy sauce fermentation enhanced the level of total free amino acids in soy sauce but reduced its glutamate level [11], which was consistent with the present observation.

As shown in Table 2, the most abundant free amino acid was lysine, followed by leucine, alanine, phenylalanine and glutamic acid in the control and sample, indicating that sonication during the enzymatic hydrolysis process only increased the contents of FAAs in EHBL but did not change their compositions of FAAs. As for taste intensity, although the variety of umami FAAs was the least, its taste intensity was relatively high due to the low thresholds of umami FAAs. Among the FAAs, the taste intensity values of glutamic acid, valine, phenylalanine and lysine in both EHBLs were greater than two, indicating that they had significant contributions to EHBL. In addition, the taste intensity values of serine, glycine, proline, arginine, isoleucine and threonine were lower than their corresponding thresholds, suggesting that these six FAAs might have slight impacts on the taste of EHBL. Although the concentration of free amino acids determined the taste of EHBL to some extent, Gao et al. found that the beef potentiators were rich in peptides with molecular weights of 1–5 kDa, which were considered as flavour enhancing peptides [1]. Therefore, peptides released during sonication and enzymatic hydrolysis might result in the differences in taste between the sample and control, which deserve further investigation in the future.

### 3.4. Analysis of Neutral Protease Activity

As shown in Figure 3, protease activities of both EHBLs gradually decreases over time except for those of EHBLs sampled at 60 min (supplementing Flavourzyme^®^ 500 MG), but protease activities of sonicated EHBL are significantly higher than those of EHBL during the whole enzymatic hydrolysis (*p* < 0.05), suggesting that appropriate sonication was capable of enhancing the protease activities during enzymatic hydrolysis. Previous studies demonstrated that low-intensity ultrasound (20 kHz, <110.6 W/cm^2^, KQ-300DE, Kunshan Ultrasonic Instrument Co., Ltd. Kunshan, China) increased the activities of neutrase and papain, and energy-gathered ultrasound (63.2 W/cm^2^, 4 min) enhanced the alcalase activity. Huang et al. concluded that ultrasound with appropriate frequency and low intensity could improve conformations of enzyme and substrate via the magnetostrictive, mechanical oscillation and cavitation actions of ultrasound, resulting in the enhancement of enzyme activity and substrate degradation rate [13]. The present result conformed to the results of previous reports.

### 3.5. Characterization of Volatile Compounds

As shown in Table 3, a total of 33 volatile compounds including alcohols, acids, ketones, aldehydes, furans, esters, alkanes and other compounds are identified in the sample and control. Among them, aldehydes and alcohols account for 72.72% of the identified volatile compounds in both EHBLs.

In this work, only hexanoic acid, a degraded product of fatty acid, was detected, which was consistent with the weak sweaty aroma in our sensory evaluation. Unsaturated fat acids can be degraded into short-chain aldehydes and unsaturated aldehydes [31,32]. Aldehydes are believed to not only contribute to the aroma of food including beef, but also generate other aroma compounds through carbonyl-amine reaction with amino acids [33]. Aldehydes dominate the aroma of enzymatically hydrolysed products due to their low odour thresholds, and they are mainly derived from the oxidative decomposition of fats and the thermally degraded products of reducing sugars and amino acids [25,34]. In this study, a total of 14 aldehydes including octanal, nonanal, decanal, etc., were identified, which were important components of EHBL aroma. Of them, nonanal had an obvious rose/citrus-like aroma, which was mainly produced by oleic acid oxidation [35]. Nonanal was reported to be a key aroma compound in roasted beef and contributed to the overall flavour of beef [36].

A total of 10 volatile alcohols, the oxidative decomposition products of fats or the reduction products of carbonyl compounds, were identified in both EHBLs [37]. 1-Pentanol, 3-ethyl-2-pentanol and 4-methyl-5-decanol were only detected in the sample, indicating that sonication promoted the oxidative decomposition of fats in EHBL. Generally, unsaturated alcohols contribute more to the aroma of food due to their lower thresholds compared to saturated alcohols. As an unsaturated alcohol, 1-octen-3-ol is a well-known compound with a strong mushroom-like aroma [38]. This compound also has fishy and fatty aromas and is produced by enzymatic or nonenzymatic degradation of linoleic acid [39]. However, it might contribute slightly to the aroma of EHBL due to its low concentration. In addition, hexanol, octanol and nonanol were reported to have grassy and fruity aromas, which were helpful for improving the aroma of food [24]. It was worth noting that some volatile substances including (E)-2-hexadecenal, 2,4-dimethylhexane, 3-hydroxy-2-butanone, 3-ethyl-2-methyl-1,3-hexadiene and 1-nonene-4-ol were only detected in the control. It was speculated that these substances contained branched and unsaturated bonds which were easily damaged by the cavitation effect of ultrasound and then degraded into other substances.

Most esters are volatile compounds with fruity aroma, which are mainly produced by esterification reaction between acids and alcohols. In this work, only diethyl phthalate with an imperceptible aroma was detected in both EHLBs, indicating its contribution the aroma of EHLB could be neglected. Furans are produced during caramelization and sugar degradation in the Maillard reaction, which were reported to be important aroma compounds in meat products due to their non-negligible effect on the overall aroma [40]. Thus, 2-pentylfuran was an important odorant detected in both EHBLs. In summary, appropriate ultrasound-assisted enzymatic hydrolysis treatment had a positive promoting effect on the odour change in EHBL.

### 3.6. Key Odour-Active Compounds Identified by OAVs

OAV is an important quantitative index to evaluate the contribution of volatile substances to food aroma [41]. In order to further clarify the influence of sonication on aroma of EHBL, OAVs of aroma-active compounds in the control and sample were measured in this study. As shown in Table 4, 13 volatile odorants including 9 aldehydes, 2 alcohols, 1 acid and 1 furan with OAV ≥ 1 in both EHBLs were identified, of them, aldehydes accounted for 69.23%, which contributed the most to EHBL aroma. As aforementioned, sonication significantly reduced the fishy odour of the sample, as shown in Table 4; the concentrations OAVs of compounds with a fishy odour, i.e., hexanoic acid and nonanal, were decreased significantly, which decreased by 35.29% and 26.03%, respectively. In addition, the flavour dilution factors of these two substances were also identified to decrease from 32 to 16. The possible reason for this phenomenon was that the fishy-odour substances were unstable. The aldehydes and the structure of double bonds were destroyed by sonication. Furthermore, some aldehydes identified, i.e., octanal, nonanal, (E)-2-octenal, (E,E)-2,4-decadien-1-al, exhibited nutty, butter, burning and fruity aromas would also contribute to the overall aroma of EHBL. Based on the OAVs and FD factors in Table 4, it can be concluded that aldehydes were the largest contributors to EHBL aroma, while sonication effectively promoted the production of most aldehydes and reduced the contents of fishy-aroma compounds, making EHBL aroma more popular.

For alcohols, only 1-hexanol and 1-octene-3-ol were identified as aroma-active compounds. After sonication, the concentration of alcohols in the sample was increased by 115.88% compared with that in the control. The concentrations and OAVs of 1-hexanol and 1-octanol in the sample increased to some extent, but the concentrations of 1-octen-3-ol in the sample and control were basically the same. Notably, the increase of 1-hexanol in the sample was more significant; the concentration of 1-hexanol in the sample was 13.4 times that in the control. As a high-molecular-weight fatty alcohol, 1-hexanol was described as having a fruity and grassy fragrance. 1-octen-3-ol was mainly formed by arachidonic acid through oxidation of lipoxygenase to form a mushroom-like odour [24]. They were important for the richness of EHBL aroma.

In addition, the OAVs of hexadecanal and (E)-2-octenal were relatively low, but they endowed EHBL with a fruity, butter, fatty and nutty aroma, making its overall aroma more harmonious. Among these key odour-active substances, octanal, (2E)-2-nonenal, trans-2-decenal and (E,E)-2,4-decadien-1-al had higher OAVs; meanwhile, they had high FD factors, suggesting they contributed greatly to the aroma of both EHBLs. It was worth noting that the trans-2-decenal FD value of ultrasound-treated beef homogenate significantly increased. Overall, the observed changes in OAVs of the major aroma-active compounds were consistent with the sensory evaluation results in Figure 2.

## 4. Conclusions

In summary, appropriate sonication (40 kHz, 80 W/L) was a feasible method to improve the flavour of EHBL via enhancing the contents of taste compounds of EHBL and removing the off-odour compounds of EHBL. Appropriate ultrasound-assisted enzymatic hydrolysis effectively removed the unstable odour substances and promoted the releases of aroma compounds and taste substances in EHBL via enhancing the activities of enzymes used. Therefore, ultrasonic technology has a great application potential in the production of EHBL with high-quality flavour. Currently, research on the biochemical mechanism of sonication improving the flavour of EHBL is ongoing.

## Figures and Tables

**Figure 1 foods-12-04460-f001:**
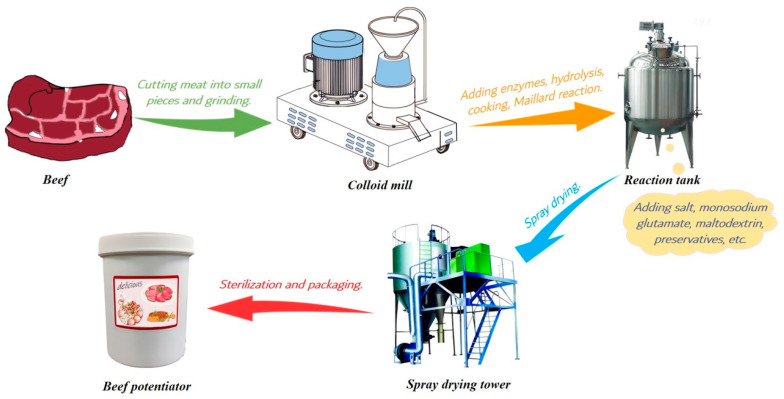
Flow chart of beef potentiator preparation.

**Figure 2 foods-12-04460-f002:**
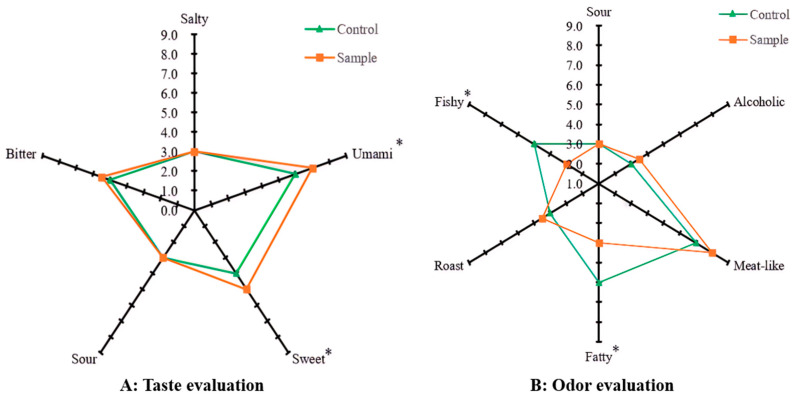
Sensory evaluation of the sample and control. * Indicates significant difference between the sample and control (*p* < 0.05).

**Figure 3 foods-12-04460-f003:**
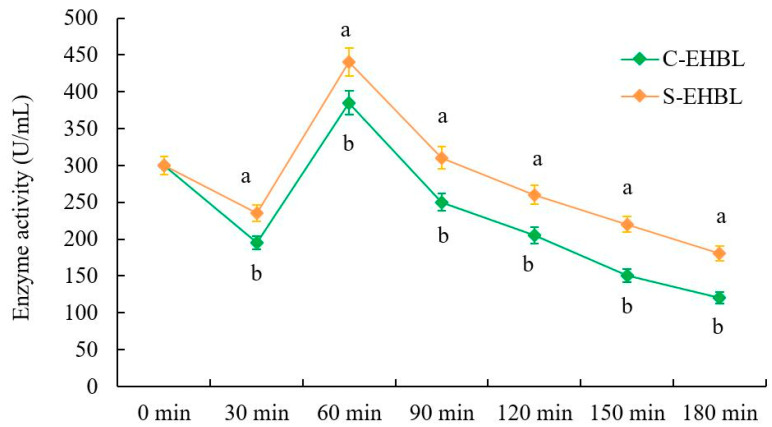
Effect of sonication on neutral protease activity of EHBL. Annotation: C-EHBL represents the control and S-EHBL represents the sample; values at the same time with different letters indicate significant differences (*p* < 0.05).

**Table 1 foods-12-04460-t001:** Physicochemical properties of the control and sample.

	Total Nitrogen (g/L)	Formaldehyde Nitrogen (g/L)	Hydrolysis Degree (%)	Total Sugars (g/L)	Reducing Sugars (g/L)
Control	8.83 ± 0.21 ^b^	1.01 ± 0.01 ^b^	16.33 ± 0.48 ^b^	0.93 ± 0.02 ^b^	0.84 ± 0.02 ^b^
Sample	10.53 ± 0.15 ^a^	1.21 ± 0.02 ^a^	19.67 ± 0.51 ^a^	1.04 ± 0.01 ^a^	0.92 ± 0.03 ^a^

Annotation: results are expressed as mean ± standard deviation (*n* = 3); values followed by different letters in the same column indicated significant differences between the data (*p* < 0.05).

**Table 2 foods-12-04460-t002:** FAA compositions of the control and sample.

FAAs	Control (g/L)	Sample (g/L)	Taste Attributes	Thresholds (g/L)	Taste Intensities	
					Control	Sample
Aspartic acid	0.14 ± 0.01 ^b^	0.25 ± 0.01 ^a^	Umami	0.17	0.82	1.47
Glutamic acid	0.43 ± 0.01 ^a^	0.40 ± 0.01 ^a^	Umami	0.075	5.73	5.33
**Umami FAAs**	**0.57**	**0.65**			**6.55**	**6.80**
Serine	0.35 ± 0.02 ^a^	0.33 ± 0.01 ^a^	Sweet	1.50	0.23	0.22
Glycine	0.48 ± 0.03 ^a^	0.52 ± 0.01 ^a^	Sweet	1.30	0.37	0.40
Threonine	0.12 ± 0.01 ^b^	0.27 ± 0.02 ^a^	Sweet	5.20	0.02	0.05
Alanine	0.75 ± 0.03 ^a^	0.73 ± 0.01 ^a^	Sweet	0.60	1.25	1.21
Proline	0.09 ± 0.01 ^b^	0.30 ± 0.01 ^a^	Sweet	3.00	0.03	0.10
Lysine	1.16 ± 0.04 ^b^	1.28 ± 0.01 ^a^	Sweet	0.50	2.32	2.56
**Sweet FAAs**	**2.95**	**3.53**			**4.22**	**4.54**
Histidine	0.16 ± 0.01 ^a^	0.16 ± 0.01 ^a^	Bitter	0.20	0.80	0.80
Arginine	0.03 ± 0.01 ^b^	0.12 ± 0.01 ^a^	Bitter	0.50	0.06	0.24
Valine	0.37 ± 0.02 ^a^	0.34 ± 0.02 ^a^	Bitter	0.10	3.70	3.40
Tyrosine	0.29 ± 0.01 ^b^	0.34 ± 0.01 ^a^	Bitter	0.46	0.63	0.74
Methionine	0.37 ± 0.02 ^a^	0.39 ± 0.02 ^a^	Bitter	0.30	1.23	1.30
Leucine	1.04 ± 0.05 ^b^	1.15 ± 0.06 ^a^	Bitter	0.95	1.09	1.21
Isoleucine	0.32 ± 0.02 ^a^	0.25 ± 0.02 ^b^	Bitter	0.90	0.36	0.28
Phenylalanine	0.47 ± 0.04 ^b^	0.59 ± 0.04 ^a^	Bitter	0.23	2.04	2.57
**Bitter FAAs**	**3.05**	**3.33**			**9.91**	**10.54**
Cysteine	0.39 ± 0.03 ^b^	0.45 ± 0.01 ^a^	Tasteless			
**Total**	**6.96**	**7.96**			**20.68**	**21.88**

Annotation: results are expressed as mean ± standard deviation (*n* = 3); values followed by different letters in the same row indicated significant differences between the data (*p* < 0.05).

**Table 3 foods-12-04460-t003:** Volatile compounds identified in the control and sample.

Compounds	CAS	RI (DB-Wax)	Aroma Description	Identification	
				Control	Sample
1-Pentanol	71-41-0	1257	Balsamic	nd	MS/RI/O
3-Ethyl-2-pentanol	609-27-8	1452	un	nd	MS/RI
1-Hexanol	111-27-3	1365	Resin, flower, green	MS/RI/O	MS/RI/O
Heptaldehyde	111-71-7	1178	Fat, citrus, rancid	MS/RI/O	MS/RI/O
(E)-Hept-2-enal	18829-55-5	1245	Soap, fat, almond	MS/RI/O	MS/RI/O
Hexanoic acid	142-62-1	1832	Sweat	MS/RI/O	MS/RI/O
Heptanol	111-70-6	1470	Fruity, green	MS/RI/O	MS/RI/O
1-Octen-3-ol	3391-86-4	1397	Mushroom, fishy, grass, fatty	MS/RI/O	MS/RI/O
2-Pentylfuran	3777-69-3	1241	Green bean, butter, pungent	MS/RI/O	MS/RI/O
Dodecane	112-40-3	1210	Alkane	MS/RI/O	MS/RI/O
Octanal	124-13-0	1282	Fat, soap, lemon, green	nd	MS/RI/O
(E)-2-Octenal	2548-87-0	1350	Meaty, fatty, green, nut, fat	MS/RI/O	MS/RI/O
1-Octanol	111-87-5	1555	Fruity	MS/RI/O	MS/RI/O
4-Methyl-5-decanol	213547-15-0	1955	un	nd	MS/RI
1-Nonanal	124-19-6	1387	Fat, citrus	MS/RI/O	MS/RI/O
(2E)-2-Nonenal	18829-56-6	1530	Cucumber, fat, green	MS/RI/O	MS/RI/O
1-Nonano	143-08-8	1504	Fat, green	MS/RI/O	MS/RI/O
Decanal	112-31-2	1485	Soap, orange peel, tallow	MS/RI/O	MS/RI/O
Trans-2-Decenal	3913-81-3	1592	Tallow, mushroom	MS/RI/O	MS/RI/O
Nonadecane	629-92-5	1911	Alkane	MS/RI/O	MS/RI/O
Undecanal	112-44-7	1649	Oil, pungent, sweet	MS/RI/O	MS/RI/O
(E,E)-2,4-Decadien-1-al	25152-84-5	1715	Fried, wax, fat	MS/RI/O	MS/RI/O
(E)-2-Hexadecenal	22644-96-8	2655	un	MS/RI	nd
Pentadecanal	2765-11-9	2060	Fresh	nd	MS/RI/O
Diethyl phthalate	84-66-2	1711	un	MS/RI/O	MS/RI/O
Docosanal	57402-36-5	2754	un	nd	MS/RI
Hexadecanal	629-80-1	2156	Strawberry and bayberry like aroma	nd	MS/RI/O
2,4-Dimethylhexane	589-43-5	820	un	MS/RI	nd
3-Hydroxy-2-butanone	513-86-0	1290	Butter, cream, milky	MS/RI/O	nd
Undecane	1120-21-4	1110	Alkane	MS/RI/O	nd
3-Ethyl-2-methyl-1,3-hexadiene	61142-36-7	1030	un	MS/RI	nd
1-Nonen-4-ol	35192-73-5	1657	un	MS/RI	nd
N-heneicosanal	51227-32-8	2846	un	MS/RI	nd

Annotation: nd: not detected in GC-MS or GC-O; un: unavailable; MS, mass spectrometry; RI, retention index; O, olfactometry.

**Table 4 foods-12-04460-t004:** Odour-active compounds identified in the control and sample.

No.	Compounds	Threshold (μg/kg)	Concentration (μg/kg)		OAV		FD	
			Control	Sample	Control	Sample	Control	Sample
1	Octanal	0.70	50.25 ± 2.65 ^a^	48.55 ± 1.35 ^b^	71.79	69.36	32	32
2	Hexanoic acid	0.08	1.87 ± 0.11 ^a^	1.21 ± 0.10 ^b^	23.38	15.13	32	16
3	1-Hexanol	5.60	3.63 ± 0.1 ^b^	41.32 ± 2.25 ^a^	0.65	7.38	—	4
4	Heptaldehyde	2.80	25.02 ± 1.32 ^b^	32.90 ± 2.51 ^a^	8.94	11.75	4	4
5	Nonanal	1.00	78.65 ± 2.3 ^a^	58.18 ± 2.1 ^b^	78.65	58.18	32	16
6	Decanal	0.10	2.59 ± 0.04 ^a^	2.78 ± 0.05 ^a^	25.9	27.8	16	16
7	Hexadecanal	0.91	nd	2.13 ± 0.11	nd	2.34	nd	1
8	2-Pentylfuran	5.80	12.56 ± 1.51 ^a^	10.60 ± 1.41 ^a^	2.17	1.83	1	1
9	1-Octen-3-ol	1.50	28.55 ± 1.08 ^a^	28.15 ± 0.95 ^a^	19.03	18.77	8	8
10	(E)-2-Octenal	3.00	14.96 ± 0.21 ^a^	14.36 ± 0.18 ^a^	4.99	4.79	2	2
11	(2E)-2-Nonenal	0.08	18.54 ± 1.4 ^a^	20.02 ± 1.5 ^a^	231.75	250.25	128	128
12	(E,E)-2,4-Decadien-1-al	0.07	7.56 ± 0.2 ^a^	8.11 ± 0.3 ^a^	108.00	115.86	64	64
13	trans-2-Decenal	0.30	27.54 ± 1.8 ^b^	31.25 ± 2.2 ^a^	91.80	104.17	32	64

Annotation: nd: not detected in GC-MS or GC-O; ^—^ the odour was not perceived using GC-O on DB-Wax column; for the concentrations of odour-active compounds, results are expressed as mean ± standard deviation (*n* = 3); values followed by different letters in the same row indicated significant differences between the data (*p* < 0.05).

## Data Availability

Data are contained within the article.

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
