# Peer review of "Improving the Flavour of Enzymatically Hydrolysed Beef Liquid by Sonication"

_foods, 2023, doi:10.3390/foods12244460_

Round 1
Reviewer 1 Report (New Reviewer)
Comments and Suggestions for Authors
I reviewed the manuscript title “Improving flavor of enzymatically hydrolyzed beef liquid by sonication”
The manuscript is well written with an approach to improve the flavor profile of beef. The introduction is appropriate and addresses the need of conducting this research. In my opinion, this manuscript can be considered for possible publication consideration after addressing suggestions below
Abstract
Background of the research should be revised. Research gap should be addressed clearly
Keywords: most of the keywords are already in the title. I suggest you to change to more relevant keywords
Line 62: . For example, Yang et al. (2020).. this wrong citation format. Kindly revise it
Line 83: therefore should be introduced before “the objective”
Line 97: What is Mixed beef ?
Provide citation for 2.3. Sensory Evaluation
Total sugar and reducing sugar can be revised as total sugars and reducing sugars throughout the manuscript
2.4. Analyses of Total nitrogen, Formaldehyde Nitrogen, Hydrolysis Degree, Total Sugar and 125 Reducing Sugar: authors should mention the unit of each result
2.6. Thresholds and Taste Intensity of FAAs: please mention the equation number
Line 151: …control and sample, the activity of neutral protea…. I think it's full stop instead of comma after the word sample.
2.10. Qualitative Analysis of Volatile Compounds: provide citation
2.11. Quantitative Analysis of Key Odorants and 2.12. Odor Activity Value (OAV): provide citation
Table 1: it should be total sugars and reducing sugars
Figure 3: please denote the statistical different and their Annotation
Improve the quality of Figures 1 and 2. The font is not readable
References need to be cross-checked for consistent format
Author Response
Please find the file: response to reviewer 1

Reviewer 2 Report (New Reviewer)
Comments and Suggestions for Authors
This study investigated the effect of sonication on the flavour and odour of enzymatically hydrolyzed beef liquid.
The subject of the article is interesting and falls within the scope of the Foods journal.
I wonder why the authors didn't perform sensory analysis at least twice, especially since this is one of the goals of the research.
However, some clarification and improvements should be made before acceptance.
In my opinion, in the introduction section, it could be valuable to add some references about the effect of sonication on meat (especially beef) matrix.

I am not a native English speaker, but comparing the language of this publication to the level of the Foods journal and other top journals, I believe that the article should be carefully analyzed in terms of grammar as well as vocabulary typical of a narrow research field.
Author Response
Please find the file: response to reviewer 2.

Reviewer 3 Report (New Reviewer)
Comments and Suggestions for Authors
The paper analyses the effect of ultrasound application in the enzymatic hydrolysis of beef homogenate to produce a beef potentiator. The objective and the methodology are adequate, although a better explanation of some techniques should be appreciated. Only, one sample is compared against a control but the conducted analysis seems to be appropriate. My main concern is that I miss a better explanation of the sensory results according to the volatile analysis and FFA. The differences observed in the analyses of FFA and volatiles are enough to support the differences sensorially found? I think it is necessary to include a global discussion of the data, beyond section discussions. Sometimes the text is only a description of the table but with little analysis or discussion. Maybe some statistical procedure like principal component analysis or cluster could be helpful to establish the relationships between sensorial and analytical results. I also have some other comments
Line 43: Why is it necessary to add amino acids? Are not provided by beef?
Line 47: Which are the usual enzymes involved in the process?
Line 102: Is there any reason to use neutras or flavourzyme, or any followed reference?
Line 128: if official methods (AOAC) were used for protein, formaldehyde, total sugar, and reducing sugars, it should be better to indicate.
Line 140: A description of the equipment used should be included.
Line 147: how the FAA threshold in water is quantified? The methods should be described enough to allow the experiment reproducibility.
152: please describe briefly the basics of the the technique.
Line 158: the authors describe appropriately how was done the extraction of volatile compounds but in the olfactometry, qualitative, and quantitative analyses the authors do not indicate how the volatiles are released from SPME fiber, and the conditions used to do it. Please describe.
Line 199: Figure 2 should appear after the first time it has been cited in the text, not at the end.
Line 200-205: It would be helpful to include the standard deviation in the graphs or a table with the significant differences since only means are represented in Figure 2.
Line 246: glutamic acid is related to umami taste. This fact should be stressed in the text.
Line 248: Which are the amino acids related to sweet, bitter, and umami? Please include this information in the text.
Line 260: The taste intensity of bitter-free amino as a whole is higher than umami, What does it mean in terms of flavor? Although there are significant differences between sample and control in each FAA, when total umami, sweet, and bitter FAA are compared the differences between samples and control are not so drastic. Although authors have analyzed free amino acids, also peptides can be released during hydrolysis contributing to flavor. I would recommend to consider this point in the discussion.
Line 314: How could be explained that if the presence of 1-octen-3-ol is associated with a fishy odor and it was present at similar concentrations in the sample and control, there was less fishy odor in the sample than in the control?
Line 355: It seems that the higher differences between sample and control were in 1-hexanol, nonanal, nonenal and hexanoic acid. Could this be related to specific flavor differences? Please consider this in the discussion.
Line 381: Which are the odor substances that have been removed by ultrasound in the sample? This fact has not been described and discussed in the text.
Comments on the Quality of English Language
The English grammar and style are correct
Author Response
Please find the file: response to reviewer 3.

This manuscript is a resubmission of an earlier submission. The following is a list of the peer review reports and author responses from that submission.
Round 1
Reviewer 1 Report
Comments and Suggestions for Authors
The manuscript entitled: "Improving flavor of enzymatically hydrolyzed beef liquid by sonication" is about the application of sonication to improve falvor of beef liquid. In general, the research is interesting, well-designed and fits the journal's aims and scopes. Some comments to improve the quality of the manuscript are as follows:
1- Title: Recommend revising the title and making it neutral.
2- Abstract: it is a good abstract, make it shorter.
3- Keywords: "enzymatically hydrolyzed beef liquid" it is a long keyword, choose keywords other than main words in the title.
4- Introduction, The statement of the problem is not clear, improve the literature and clearly mention the originality of your work.
5- Methods: Clear and lots of details, make it shorter and remove unnecessary details.
6- Results and discussions: This part is OK.
7- Conclusion: Add a few future research recommendations.
8- References: Use the most recent publications and replace those belong outdated if possible.